# Temporal Trends in HIV-1 Mutations Used for the Surveillance of Transmitted Drug Resistance

**DOI:** 10.3390/v13050879

**Published:** 2021-05-11

**Authors:** Soo-Yon Rhee, Philip L. Tzou, Robert W. Shafer

**Affiliations:** Department of Medicine, Stanford University, Stanford, CA 94305, USA; philiptz@stanford.edu (P.L.T.); rshafer@stanford.edu (R.W.S.)

**Keywords:** HIV-1, antiretroviral therapy, drug resistance, surveillance

## Abstract

In 2009, a list of nonpolymorphic HIV-1 drug resistance mutations (DRMs), called surveillance DRMs (SDRMs), was created to monitor transmitted drug resistance (TDR). Since 2009, TDR increased and antiretroviral therapy (ART) practices changed. We examined the changing prevalence of SDRMs and identified candidate SDRMs defined as nonpolymorphic DRMs present on ≥ 1 expert DRM list and in ≥0.1% of ART-experienced persons. Candidate DRMs were further characterized according to their association with antiretrovirals and changing prevalence. Among NRTI-SDRMs, tenofovir-associated mutations increased in prevalence while thymidine analog mutations decreased in prevalence. Among candidate NRTI-SDRMs, there were six tenofovir-associated mutations including three which increased in prevalence (K65N, T69deletion, K70G/N/Q/T). Among candidate NNRTI-SDRMs, six that increased in prevalence were associated with rilpivirine (E138K/Q, V179L, H221Y) or doravirine (F227C/L) resistance. With the notable exceptions of I47A and I50L, most PI-SDRMs decreased in prevalence. Three candidate PI-SDRMs were accessory darunavir-resistance mutations (L10F, T74P, L89V). Adding the candidate SDRMs listed above was estimated to increase NRTI, NNRTI, and PI TDR prevalence by 0.1%, 0.3%, and 0.3%, respectively. We describe trends in the prevalence of nonpolymorphic HIV-1 DRMs in ART-experienced persons. These data should be considered in decisions regarding SDRM list updates and TDR monitoring.

## 1. Introduction

In 2009, an expert panel developed a list of HIV-1 drug resistance mutations (DRMs) for the purpose of monitoring transmitted drug resistance (TDR). The list was based on three main criteria. The first was that the DRMs should be recognized as causing or contributing to drug resistance—defined as being present multiple expert lists of DRMs. The second was that the DRMs should be nonpolymorphic (i.e., not occur in the absence of selective drug pressure) regardless of HIV-1 subtype. The third was that the DRM list should be parsimonious. As its purpose was for surveillance, rather than clinical care, the list would not need to include exceedingly rare DRMs.

The published list was widely adopted. It has been used by the WHO and many research laboratories to calculate TDR prevalence in many surveillance studies [1,2]. This list of “surveillance” DRMs (SDRMs) contained 93 DRMs including 34 NRTI-associated DRMs at 15 RT positions, 19 NNRTI-associated DRMs at 10 RT positions and 40 PI-associated DRMs at 18 positions. However, more than 10 years have elapsed since the SDRM list was developed. Antiretroviral therapy (ART) guidelines and treatment practices have changed, and several new antiretroviral drugs have been introduced. In a recent study, we and others reported the prevalence of integrase strand transfer inhibitor (INSTI)-resistance associated mutations in INSTI-naïve and INSTI-experienced persons and developed a list of nonpolymorphic INSTI-selected DRMs that could be useful for the surveillance of INSTI-TDR [3]. Therefore, we sought to determine how well the original SDRM list met the original selection criteria and whether any additional DRMs might be useful as part of an expanded list to monitor TDR.

We examined the prevalence of each SDRM in ART-naïve and ART-experienced persons in the years before and after 2009. The ART-naïve prevalence was examined in each of the most common subtypes and circulating recombinant forms. To identify potential additions to the SDRM list, which we called candidate SDRMs, we performed similar analyses to those that we performed to select the initial SDRM list. Candidate SDRMs that were present on multiple lists, which increased in prevalence in ART-experienced persons, and/or were associated with reduced susceptibility to new antiretroviral drugs or drugs used with increasing frequency were considered to be potentially useful additions to expanded SDRM list. This study, however, does not recommend specific changes to the SDRM list as such a change would require additional expert input.

## 2. Materials and Methods

### 2.1. Lists of Mutations

The first step in developing the 2009 SDRM list, was to pool all mutations present on three or more of the following five expert mutation lists: ANRS drug resistance interpretation algorithm (July 2008), HIVdb drug resistance interpretation algorithm (4.3.7), IAS-USA Mutations Associated with Drug Resistance (March/April 2008), Los Alamos National Laboratories HIV Sequence database (2007), or Rega Institute Drug Resistance Interpretation Algorithm (7.1.1). In 2008 the LANL HIV Sequence Database discontinued their drug resistance mutation list and the remaining four lists gradually evolved. Therefore, in this analysis, we used the remaining four lists of mutations as the starting point for our analyses. As of 1 March 2020, these four lists contained a total of 113 PI-, 66 NRTI-, and 69 NNRTI-associated mutations (Appendix A). All 2009 SDRMs were included among these mutations. The remaining mutations were defined as non-SDRMs.

### 2.2. Numbers of Sequences

The analysis that led to the 2009 SDRM list included four sets of quality-controlled sequences including 15,117 PR sequences from PI-naïve persons, 11,586 RT sequences from RTI-naïve persons, 7819 PR sequences from PI-experienced persons, and 14,622 RT sequences from RTI-experienced persons. More than half of the ART-class naïve sequences were from one of the seven most common non-B subtypes including > 1000 sequences belonging to subtypes A, C, and CRF02_AG and > 300 sequences belonging to subtypes D, F, G, and CRF01_AE. Approximately one-third of the RTI-experienced and one-fifth of the PI-experienced sequences belonged to a non-B subtype.

The analysis performed for this study includes quality-controlled 115,880 PR sequences from PI-naïve persons, 100,850 RT sequences from RTI-naïve persons, 32,800 PR sequences from PI-experienced persons and 56,573 RT sequences from RTI-experienced persons. The PR sequences from PI-naïve persons included 61,313 non-B sequences including > 5000 sequences belonging to subtypes A, C, CRF01_AE, and CRF02_AG and >1000 sequences belonging to subtypes D, F and G. The non-B PR sequences also included about 4500 sequences belonging to other CRFs and unique recombinant forms (Appendix A). The RT sequences from RTI-naïve persons included 51,575 non-B sequences including >5000 belonging to subtypes A, C, CRF01_AE, and CRF02_AG and >1000 sequences belonging to subtypes D, F and G. The non-B RT sequences also included about 8400 sequences belonging to other CRFs and unique recombinant forms (Appendix A). Quality control involved restricting our analysis to sequences generated by direct PCR dideoxy-nucleotide sequencing from plasma samples that displayed no evidence for G-to-A hypermutation or an excess of highly unusual mutation or stop codons.

### 2.3. Analysis

In our 2009 analysis, two steps were taken to reduce the influence of transmitted resistance on the identification of nonpolymorphic mutations [4]. First, we excluded sequences with two or more mutations from an earlier list of SDRMs published in 2007 [5] based on the premise that such sequences were likely to have resulted from previous selective drug pressure. Second, we excluded sequences from individuals residing in areas with high levels of TDR. For our current analysis, we adopted the first step. However, because TDR has become common in many regions, we have not excluded sequences from any geographic region. Nonpolymorphic DRMs were then defined as those with an overall prevalence below 0.5% in ART-class naïve persons and below 1.0% in each of the eight major subtypes circulating recombinant forms (CRFs): A, B, C, D, F, G, CRF01_AE, and CRF02_AG in ART-class naïve persons.

Our analyses involved determining the prevalence of each SDRM and each non-SDRM among ART-class naïve and ART-experienced persons before and after 2009. For each SDRM and each non-SDRM, we determined the mutation’s overall prevalence and its prevalence in each of the eight most common subtypes/CRFs. The highest prevalence of a mutation in the major subtypes was called its maximum-subtype prevalence. The prevalence of a mutation in treated persons divided by its prevalence in naïve persons was referred to as its treatment/naïve prevalence ratio. To quantify the change in prevalence of DRMs before and after 2009, we compared the prevalence of each DRM after 2009 with the prevalence before 2009 using Fisher’s exact test. *p* values were adjusted using Holm’s method to control the family-wise error rate for multiple hypothesis testing. The changes in the prevalence with an adjusted *p* values < 0.05 were considered significant and ranked according to the odds ratios (ORs). To evaluate the potential impact of adding mutations to the SDRM list, we selected recently published TDR studies that contained sequences from ≥1000 ART-naïve persons (Appendix A) [6].

Candidate SDRMs were defined as nonpolymorphic non-SDRMs present on three or more expert mutation lists that had a prevalence ≥ 0.1% in treated persons since 2009. The requirement for a prevalence of ≥ 0.1% in treated persons was to avoid the consideration of exceedingly rare DRMs. Thymidine-analog mutations were defined as the NRTI-SDRMs M41L, D67N/G/E, K70R, L210W, T215Y/F/S/C/D/E/I/V, and K219Q/E/N/R. T215 mutations other than T215Y/F were called T215 revertants because they often emerge in individuals initially infected with a virus containing T215Y/F [7,8]. Tenofovir-associated mutations were defined as A62V, K65R/N/E, S68G/N/D, T69 deletions, and K70E/Q/N/T/S/G [9].

## 3. Results

### 3.1. NRTI Mutations

The prevalence of each NRTI-SDRM in RTI-naïve and NRTI-experienced persons before and after 2009 is shown in Table 1. Five SDRMs increased in prevalence among NRTI-experienced persons including K65R, K70E, Y115F, and M184V/I. Twenty-five NRTI-SDRMs decreased in prevalence among NRTI-experienced persons; four NRTI-SDRMs demonstrated non-statistically significant decreases in prevalence. Three mutations demonstrated a change in prevalence among RTI-naïve persons: M41L, F77L, and T215D each decreased in prevalence. Twenty-six (76.4%) of the 34 NRTI-SDRMs had a prevalence of ≥ 1.0% among NRTI-experienced persons since 2009. Not surprisingly, four of the T215 revertants have low treated/naïve prevalence ratios ranging from 1-fold to 11-fold prior to 2009 and from 1-fold to 6-fold since 2009. Appendix A (A) displays the yearly prevalence of each NRTI-associated SDRM in NRTI-experienced persons.

Among the 32 NRTI-associated non-SDRMs, 17 were considered candidate SDRMs because they were nonpolymorphic and had a treatment prevalence ≥ 0.1% since 2009 (Table 2). Three of these mutations increased and eight decreased in prevalence in NRTI-experienced persons since 2009. There were no changes in the prevalence of any of these mutations in RTI-naïve persons. The 17 candidate SDRMs had a prevalence between 0.13% and 0.86% among NRTI-experienced persons. Three of the mutations that did not change in prevalence had very low treatment/naïve prevalence ratios. Appendix A (A) displays the yearly prevalence of each NRTI-associated candidate SDRM in NRTI-experienced persons. Appendix A contains the ORs and adjusted *p* values comparing the prevalence of each NRTI-associated SDRM and candidate SDRM before and after 2009 in RTI-naïve and NRTI-experienced persons.

The three candidate NRTI-SDRMs (T69deletion and K70Q/T) that increased in prevalence among NRTI-experienced persons were tenofovir-associated mutations [9,10]. Three additional tenofovir-associated mutations (K65N and K70G/N) did not increase in prevalence [9,11,12]. These six tenofovir-associated mutations had a treatment prevalence ranging from 0.17% to 0.73%. With the exception of K65N, which was on four expert lists, the remaining tenofovir-associated mutations were on just one or two lists. These six candidate SDRMs remain rare in RTI-naïve persons and their addition to the current list of NRTI-SDRMs would have increased the prevalence of NRTI-associated TDR by a median of just 0.1% (range: 0 to 0.2%) in the representative sample of 10 recent TDR studies (Appendix A). Figure 1 shows the locations of the NRTI-SDRMs and the six candidate mutations (K65N, T69deletion and K70G/N/Q/T) within the three-dimensional structure of the polymerase coding region of p66 HIV-1 RT.

A62V, another tenofovir-associated mutation, was not considered a candidate SDRM because it had a prevalence of 6.5% in subtype A sequences owing to a founder effect among subtype A6 sequences, one of the dominant variants in countries of the former Soviet Union. A62V had a naïve prevalence below 0.2% in each of the other subtypes and increased in prevalence from 4.1 to 6.9% since 2009.

### 3.2. NNRTI Mutations

The prevalence of each NNRTI-SDRM in RTI-naïve and NNRTI-experienced persons before and after 2009 is shown in Table 3. Seven SDRMS increased in prevalence among NNRTI-experienced persons including K101E, Y181V, Y188C, G190S, P225H, and M230L while four decreased in prevalence including L100I, V106A, K101P, and V181I. V106M displayed the largest increase in prevalence, increasing from 2.6 to 9.1%. Four SDRMs increased in prevalence in RTI-naïve persons including K103N, V106M, and G190A/E while L100I decreased in prevalence among RTI-naïve persons. Fourteen (73.7%) of the 19 NNRTI-SDRMs had a prevalence ≥ 1.0% among NNRTI-experienced persons since 2009. The SDRMs with the lowest treated/naïve prevalence ratios were G190E (11-fold) and K103N (22-fold). Appendix A (B) displays the yearly prevalence of each NNRTI-associated SDRM in NNRTI-experienced persons.

Among the 50 NNRTI-associated non-SDRMs, 18 were considered candidate SDRMs because they were nonpolymorphic and had a treatment prevalence ≥ 0.1% since 2009 (Table 4). Seven of these DRMs increased in prevalence among NNRTI-experienced persons since 2009 including E138K/Q, V179L/M, H221Y, and F227C/L. Each of these seven, with the exception of V179M were on three or more expert mutation lists. The prevalence of these seven mutations in NNRTI-experienced persons ranged from 0.1% for F227C to 8.1% for H221Y. Two of these mutations had relatively low treated/naïve prevalence ratios including E138K (6-fold) and V179M (11-fold). Appendix A(B) displays the yearly prevalence of each NNRTI-associated candidate SDRM in NNRTI-experienced persons. Appendix A contains the ORs and adjusted *p* values comparing the prevalence of each NNRTI-associated SDRM and candidate SDRM before and after 2009 in RTI-naïve and NNRTI-experienced persons.

Six of the candidate NNRTI-SDRMs that increased in prevalence are associated with reduced susceptibility to rilpivirine (E138K/Q, V179L, and H221Y) [13,14] or doravirine (F227C/L) [15,16,17] and are present on three or more of the 2020 expert mutation lists. The addition of these six mutations the current list of NNRTI-SDRMs would have increased the prevalence of NNRTI-associated TDR by a median of 0.3% (range: 0 to 0.6%) in the representative sample of 10 recent TDR studies (Appendix A). Figure 2 shows the locations of the NNRTI-SDRMs and the six candidate mutations (E138K/Q, V179L, H221Y, and F227C/L) within the three-dimensional structure of the polymerase coding region of p66 and the finger domain of p51 HIV-1 RT.

E138R and M230I are not present in Table 4 because they occurred in fewer than 0.1% of NNRTI-experienced persons [14,18]. M230I also had a low treated/naïve ratio of 4. Despite being associated with high-level resistance to each of the NNRTIs [16,18] G190E is uncommon in NNRTI-experienced persons. Its overall low prevalence is likely related to its reduced replication capacity [19]. Its occasional occurrence in naïve individuals may result from the fact that it can be caused by APOBEC. Although we excluded sequences meeting previously defined criteria for APOBEC-mediated G-to-A hypermutation [20], we cannot exclude the possibility that some of the G190E mutations were APOBEC-mediated. A similar phenomenon may also explain the relatively low treated/naïve prevalence ratios for E138K and M230I.

### 3.3. PI Mutations

The prevalence of each PI-SDRM in PI-naïve and PI-experienced persons before and after 2009 is shown in Table 5. Two SDRMS increased in prevalence among PI-experienced persons including V47A and I50L while 34 decreased in prevalence. N88S decreased in prevalence among PI-naïve persons while no other mutations changed in prevalence among PI-naïve persons. Twenty-two (55%) of the 40 PI-SDRMs had a prevalence ≥ 1.0% among PI-experienced persons since 2009. The PI-SDRMs with the lowest treatment/naïve prevalence ratios were F53Y (5-fold), V82L (7-fold), M46L (9-fold), and I85V (14-fold). Appendix A(C) displays the yearly prevalence of each PI-associated SDRM in PI-experienced persons.

Among the 73 PI-associated non-SDRMs, 18 were considered candidate SDRMs because they were nonpolymorphic and had a treatment prevalence ≥ 0.1% in the period since 2009 (Table 6). One of these DRMs (L89T) increased in prevalence among PI-experienced persons while 14 decreased in prevalence among PI-experienced persons. Few data are available linking L89T to reduced susceptibility to specific PIs [21]. The DRMs with the highest prevalence since 2009 included L10F (6.7%), K20T (4.1%), and K43T (3.0%). Six mutations had very low treatment/naïve prevalence ratios. Appendix A(C) displays the yearly prevalence of each PI-associated candidate SDRM in PI-experienced persons. Appendix A contains the ORs and adjusted *p* values comparing the prevalence of each PI-associated SDRM and candidate SDRM before and after 2009 in PI-naïve and PI-experienced persons.

Three of the mutations were on four expert lists including L10F, T74P, and L89V. T74P and L89V were considered to be accessory mutations based on analyses of phenotypic and clinical data derived from the POWER studies that led the FDA approval of darunavir [22]. L10F has been shown to be an accessory DRM associated with reduced in vitro lopinavir and darunavir susceptibility [23,24,25]. The addition of these three mutations the current list of PI-SDRMs would have increased the prevalence of PI-associated TDR by a median of 0.3% (range: 0 to 0.5%) in the representative sample of 10 recent TDR studies (Appendix A). Figure 3 shows the locations of the PI-SDRMs and the three candidate mutations (L10F, T74P, and L89V) within the three-dimensional structure of HIV-1 protease.

## 4. Discussion

Our analysis of NRTI-SDRMs found that there was a decreasing prevalence of thymidine analog mutations and an increasing prevalence of tenofovir-associated mutations among NRTI-experienced persons. However, the naïve prevalence of thymidine analog mutations did not decrease between the two time periods and the naïve prevalence of tenofovir-associated mutations did not increase. The continued prevalence of thymidine analog mutations in RTI-naïve persons since 2009 despite the decline in thymidine analog use has been reported in several studies and reflects the fact that with the exception of T215Y/F, the thymidine analog mutations display minimally reduced replication fitness [26,27,28]. Conversely, the continued low prevalence of tenofovir-associated mutations in RTI-naïve persons reflects the reduced fitness associated with these mutations [29].

Even though they currently remain rare in RTI-naïve persons, the tenofovir-associated mutations K65N, T69deletion, and K70G/N/Q/T were identified as potentially useful additions to an expanded NRTI SDRM list, because of the importance of tenofovir for all first-line regimens. Conversely, four of the T215 revertant mutations, T215C/D/E/S had treatment/naïve prevalence ratios of about 1.0 raising the question of whether these mutations will remain useful for the ongoing surveillance of clinically meaningful transmitted NRTI resistance.

Our analysis of NNRTI-SDRMs found that there were few major changes in their overall prevalence in either RTI-naïve or NNRTI-experienced persons with the exception of V106M which became more prevalent in NNRTI-experienced persons because of the increased proportion of subtype C viruses increased from between the two time periods.

Six candidate NNRTI-SDRMs were identified as potentially useful additions to an expanded NNRTI SDRM list because they are associated with reduced susceptibility to rilpivirine (E138K/Q, V179L, and H221Y) and doravirine (F227L/C) and because they increased in prevalence since 2009 [13,14,15,16]. 

Our analyses of PI-SDRMs found that the prevalence of most PI-resistance mutations in PI-experienced persons decreased between the two time periods, a finding consistent with published trends showing the reduced emergence of PI-resistance mutations in patients experiencing virological failure on a PI-containing regimen [30,31,32]. The two SDRMs that increased in prevalence, I50L and V47A, are associated with reduced susceptibility to atazanavir and lopinavir, respectively [33,34].

Three candidate PI-SDRMs (L10F, T74P, and L89V) were identified as potentially useful additions to an expanded PI SDRM list because they are accessory darunavir-associated resistance mutations. Four PI-SDRMs (M46L, F53Y, V82L, and I85V) had relatively low treated/naïve prevalence ratios. Moreover, F53Y, V82L, and I85V have not been linked to reduced susceptibility to the currently used PIs raising the question of whether these mutations will remain useful for the ongoing surveillance of clinically meaningful transmitted PI resistance. The potential for M46I/L to occur at low levels in subtype A and CRF01_AE populations without previous PI exposure was recognized at the time the 2009 SDRM list was developed and in several subsequent analyses [35]. However, M46I remains an important DRM for lopinavir and atazanavir, although it may continue to occur at minimally elevated levels in the aforementioned subtypes.

Monitoring TDR has been used to document the prevalence and patterns of TDR strains, identify populations at risk for having TDR at time of diagnosis, and informing the choice of first-line therapy in areas where routine genotypic resistance testing is not available. The use of a standard list of SDRMs for more than 10 years has made it possible to compare temporal trends in the prevalence of resistance in different regions. However, the use of the same list despite changing ART practices carries the risk that trends associated with newly recognized DRMs may not be recognized by groups performing surveillance. To balance these competing objectives, we have identified high-priority candidate SDRMs so that these can be brought to the attention of those performing surveillance. Decisions about whether to expand the WHO list of SDRMs, however, should ultimately be made in consultation with groups performing surveillance.

## Figures and Tables

**Figure 1 viruses-13-00879-f001:**
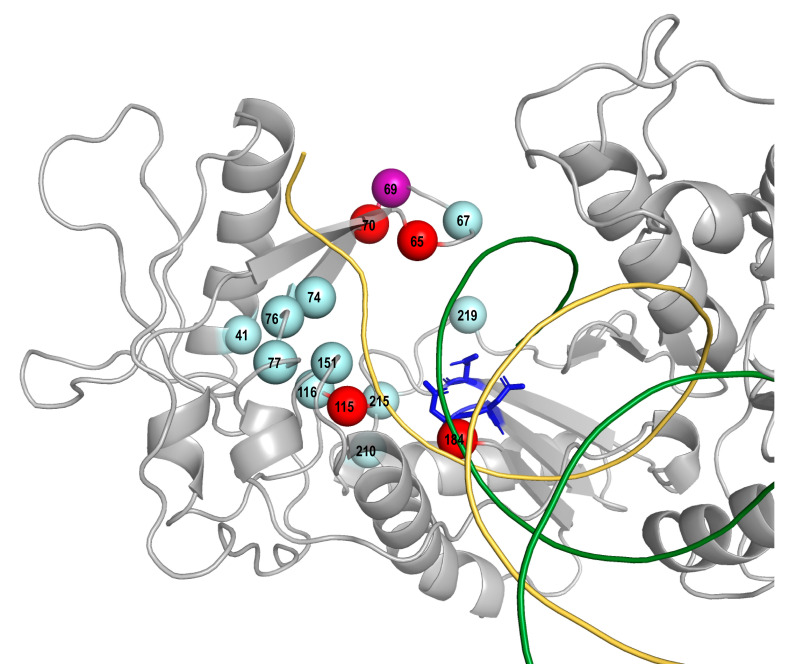
Three-dimensional structure (PDB accession number 1RTD) of HIV-1 RT showing the p66 monomer in grey. The active site residues D110, D185 and D186 are shown in stick mode in blue, a DNA template and a primer are shown in ribbon mode in yellow and green, respectively. Red spheres indicate four SDRM positions at which SDRMs including K65R, K70E, Y115F, and M184V/I and tenofovir-associated non-SDRMs (K70Q/T) increased in prevalence since 2009 in NRTI-experienced persons. A purple sphere indicates a SDRM position at which tenofovir-associated non-SDRM T69deletion increased in prevalence since 2009 in NRTI-experienced persons. Spheres in cyan indicate SDRM positions at which SDRMs decreased or did not change in prevalence since 2009 in NRTI-experienced persons.

**Figure 2 viruses-13-00879-f002:**
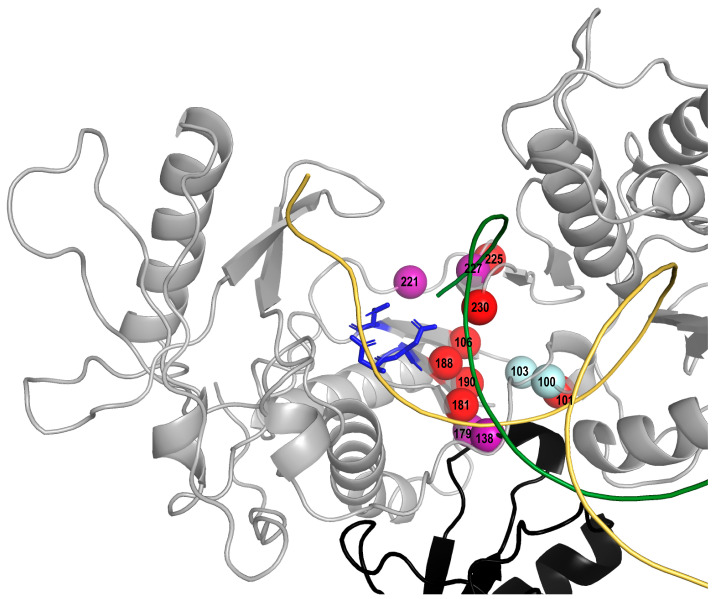
Three-dimensional structure (PDB accession number 1RTD) of HIV-1 RT showing the p66 monomer in grey and the finger domain of the p51 in black. The active site residues D110, D185, and D186 are shown in stick mode in blue, a DNA template and a primer are shown in ribbon mode in yellow and green, respectively. Red spheres indicate seven SDRM positions at which SDRMs including K101E, Y181V, Y188C, G190S, P225H, and M230L increased in prevalence since 2009 in NNRTI-experienced persons. Four spheres in purple indicate positions at which non-SDRMs (E138K/Q, V179L, H221Y, and F227C/L) increased in prevalence since 2009 in NNRTI-experienced persons. Spheres in cyan indicate two SDRM positions at which SDRMs decreased or did not change in prevalence since 2009 in NNRTI-experienced persons.

**Figure 3 viruses-13-00879-f003:**
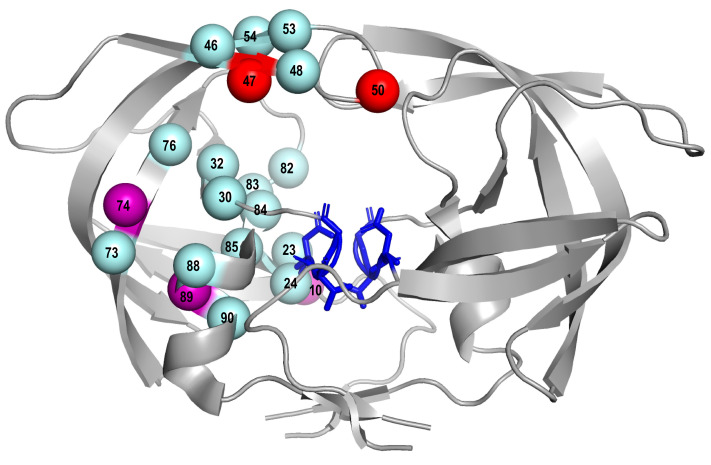
Three-dimensional structure (PDB accession number 1MUI) of HIV-1 protease showing in grey. The active site residues D25, D26, and D27 are shown in stick mode in blue. Red spheres indicate two SDRM positions at which SDRMs (V47A and I50L) increased in prevalence since 2009 in PI-experienced persons. Three purple spheres indicate darunavir-associated candidate SDRMs including L10F, T74P, and L89V. Spheres in cyan indicate SDRM positions at which SDRMs decreased or did not change in prevalence since 2009 in PI-experienced persons. HIV-1 protease is a homodimer and amino acid positions are indicated only on one monomer of the homodimer for clarity.

**Table 1 viruses-13-00879-t001:** Prevalence of Surveillance NRTI-Resistance Mutations in RTI-Naïve and NRTI-Experienced Persons before and after 2009.

				<2009			≥2009		
Pos	AA	2009	2020	%Naïve (*n* = 54,910)	%Rx (*n* = 32,363)	Rx/Naïve Ratio	%Naïve (*n* = 45,940)	%Rx (*n* = 24,210)	Rx/Naïve Ratio
**DRMs that increased in prevalence in treated persons**									
70	E	4	4	0.02 (0.23)	0.6	27	0.02 (0.04)	2.6	168
65	R	4	4	0.03 (0.07)	4.0	156	0.02 (0.04)	10.4	478
115	F	4	4	0.01 (0.10)	2.0	185	0 (0.20)	4.5	2055
184	I	4	4	0.03 (0.08)	2.3	83	0.05 (0.20)	3.3	63
184	V	4	4	0.21 (0.25)	54.5	260	0.15 (0.20)	60.6	415
**DRMs that decreased in prevalence in treated persons**									
69	ins	4	4	0 (0)	1.0	>>>	0 (0.02)	0.2	95
210	W	4	4	0.06 (0.10)	20.8	335	0.1 (0.13)	7.7	81
75	T	3	2	0 (0)	1.5	>>>	0 (0)	0.5	>>>
75	A	3	2	0.01 (0.03)	0.8	69	0.01 (0.01)	0.2	37
215	Y	4	4	0.02 (0.04)	30.5	1289	0 (0.04)	13.4	6149
41	L	4	4	0.34 (0.79)	33.0	96	0.21 (0.50)	14.8	70
67	E	2	1	0.03 (0.02)	0.6	19	0.02 (0.16)	0.2	10
219	N	3	2	0.04 (0.17)	3.4	88	0.01 (0.20)	1.4	110
215	C	3	3	0.09 (0.15)	0.9	11	0.08 (0.19)	0.4	5
151	M	4	4	0 (0)	2.9	>>>	0 (0)	1.2	>>>
69	D	3	3	0.08 (0.16)	6.9	90	0.09 (0.14)	3.1	34
116	Y	2	2	0.01 (0.03)	2.2	307	0 (0)	1.0	>>>
67	N	4	4	0.06 (0.24)	30.2	518	0.03 (0.13)	15.7	515
74	V	4	4	0.01 (0.10)	8.6	790	0 (0.04)	3.9	906
219	R	3	2	0.07 (0.08)	2.7	37	0.04 (0.17)	1.3	31
75	S	3	2	0 (0)	0.3	154	0 (0)	0.1	>>>
219	Q	4	4	0.1 (0.17)	12.6	125	0.07 (0.12)	6.9	96
70	R	4	4	0.08 (0.17)	20.6	246	0.04 (0.08)	12.4	284
215	D	3	3	0.27 (0.48)	0.8	3	0.17 (0.37)	0.4	3
215	V	3	3	0.01 (0.01)	0.9	159	0 (0.01)	0.5	114
77	L	2	3	0.15 (0.24)	1.9	12	0.07 (0.13)	1.2	19
215	F	4	4	0 (0.01)	11.8	3239	0 (0.01)	8.3	3824
215	S	3	3	0.36 (0.57)	2.5	7	0.30 (0.57)	1.8	6
74	I	3	3	0.03 (0.11)	4.6	182	0.03 (0.07)	3.7	142
219	E	4	4	0.03 (0.20)	6.8	266	0.02 (0.04)	6.1	255
**DRMs that did not change in prevalence in treated persons**									
215	I	3	3	0.03 (0.08)	2.2	70	0.03 (0.13)	2.0	62
75	M	3	2	0.03 (0.20)	3.6	116	0.05 (0.14)	3.5	69
67	G	3	2	0.03 (0.05)	2.4	71	0.02 (0.16)	2.3	130
215	E	3	3	0.12 (0.22)	0.2	1	0.15 (0.37)	0.1	1

Abbreviation: Pos—amino acid position; AA—amino acid; 2009—Number of expert lists with the mutation in 2009; 2020—Number of expert lists with the mutation in 2020; %Naïve—percent prevalence among RTI-naïve persons; %Rx—percent prevalence among NRTI-experienced persons; DRMs—drug resistance mutations. Fisher’s exact test was used to compare the prevalence of each DRM after 2009 with the prevalence before 2009 and to calculate odds ratio (OR). OR > 1 and OR < 1 with an adjusted *p* value < 0.05 using Holm’s method were considered increase and decrease in prevalence, respectively. Within each section mutations are ordered according to the OR of their change within treated persons. Three mutations decreased in frequency among RTI-naïve persons: M41L, F77L, and T215D.

**Table 2 viruses-13-00879-t002:** Prevalence of Additional Nonpolymorphic NRTI-Resistance Mutations in RTI-Naïve and NRTI-Experienced Persons before and after 2009.

				<2009			≥2009		
Pos	AA	2009	2020	%Naïve (*n* = 54,910)	%Rx (*n* = 32,363)	Rx/Naïve Ratio	%Naïve (*n* = 45,940)	%Rx (*n* = 24,210)	Rx/Naïve Ratio
**DRMs that increased in prevalence in treated persons**									
70	Q	0	1	0.02 (0.05)	0.21	10	0 (0.04)	0.58	133
69	del	1	1	0 (0)	0.13	71	0 (0)	0.30	100
70	T	1	1	0.03 (0.17)	0.32	10	0.03 (0.05)	0.73	22
**DRMs that decreased in prevalence in treated persons**									
44	A	3	1	0 (0.08)	1.74	955	0.01 (0.02)	0.59	90
219	H	1	1	0 (0)	0.34	100	0 (0)	0.14	100
215	N	2	3	0.02 (0.03)	1.33	73	0.02 (0.04)	0.73	42
67	H	1	1	0 (0)	0.23	127	0.01 (0.02)	0.13	20
69	G	2	1	0 (0.02)	0.27	146	0 (0)	0.15	100
67	S	1	1	0 (0.01)	0.35	95	0 (0)	0.20	100
219	W	1	1	0 (0)	0.28	100	0 (0)	0.17	100
69	A	3	1	0.22 (0.50)	1.09	5	0.28 (0.39)	0.86	3
**DRMs that did not change in prevalence in treated persons**									
65	N	2	4	0.02 (0.03)	0.10	4	0.01 (0.08)	0.17	26
70	N	1	1	0.04 (0.07)	0.31	8	0.03 (0.13)	0.40	12
215	A	2	3	0.12 (0.20)	0.31	3	0.1 (0.86)	0.28	3
215	L	2	3	0.06 (0.10)	0.19	3	0.06 (0.15)	0.16	3
70	G	2	2	0 (0)	0.47	100	0 (0)	0.33	100
67	T	1	1	0 (0)	0.24	100	0 (0)	0.15	100

Abbreviation: Pos—amino acid position; AA—amino acid; 2009—Number of expert lists with the mutation in 2009; 2020—Number of expert lists with the mutation in 2020; %Naïve—percent prevalence among RTI-naïve persons; %Rx—percent prevalence among NRTI-experienced persons; DRMs—drug resistance mutations. Fisher’s exact test was used to compare the prevalence of each DRM after 2009 with the prevalence before 2009 and to calculate odds ratio (OR). OR > 1 and OR < 1 with an adjusted *p* value < 0.05 using Holm’s method were considered increase and decrease in prevalence, respectively. Within each section mutations are ordered according to the OR of their change within treated persons. None of the mutations changed in prevalence among RTI-naïve persons.

**Table 3 viruses-13-00879-t003:** Prevalence of Surveillance NNRTI-Resistance Mutations in RTI-Naïve and NNRTI-Experienced Persons before and after 2009.

				<2009			≥2009		
Pos	AA	2009	2020	%Naïve (*n* = 54,910)	%Rx (*n* = 27,755)	Rx/Naïve Ratio	%Naïve (*n* = 45,940)	%Rx (*n* = 22,219)	Rx/Naïve Ratio
**DRMs that increased in prevalence in treated persons**									
106	M	4	4	0.01 (0.04)	2.6	203	0.05 (0.20)	9.1	182
225	H	4	4	0.02 (0.11)	4.3	176	0.01 (0.02)	7.4	843
230	L	3	4	0.01 (0.05)	1.5	116	0.02 (0.05)	2.6	165
188	C	4	4	0.02 (0.03)	0.7	31	0.02 (0.08)	1.0	55
181	V	4	4	0 (0)	0.5	100	0 (0.04)	0.7	337
101	E	4	4	0.14 (0.39)	7.2	51	0.19 (0.81)	9.3	49
190	S	4	4	0.01 (0.02)	2.7	369	0.02 (0.16)	3.3	190
**DRMs that decreased in prevalence in treated persons**									
181	I	4	4	0 (0.03)	0.8	223	0.01 (0.08)	0.6	68
106	A	4	4	0.03 (0.10)	1.8	70	0.01 (0.04)	1.4	103
100	I	4	4	0.03 (0.23)	4.5	155	0 (0.02)	3.5	804
101	P	4	4	0 (0.01)	1.7	936	0 (0.01)	1.4	633
**DRMs that did not change in prevalence in treated persons**									
190	E	3	4	0.02 (0.09)	0.5	25	0.05 (0.06)	0.6	11
103	S	3	4	0.03 (0.17)	2.0	69	0.05 (0.21)	2.3	43
181	C	4	4	0.16 (0.32)	17.9	112	0.17 (0.65)	18.1	108
190	A	4	4	0.16 (0.23)	14.5	93	0.24 (0.36)	14.6	60
103	N	4	4	1.32 (1.81)	35.8	27	1.64 (3.03)	35.6	22
188	L	4	4	0.05 (0.30)	4.5	95	0.07 (0.49)	4.4	60
188	H	4	4	0.03 (0.08)	0.8	28	0.01 (0.04)	0.7	77
179	F	4	3	0.01 (0.02)	0.3	18	0 (0.01)	0.2	44

Abbreviation: Pos—amino acid position; AA—amino acid; 2009—Number of expert lists with the mutation in 2009; 2020—Number of expert lists with the mutation in 2020; %Naïve—percent prevalence among RTI-naïve persons; %Rx—percent prevalence among NNRTI-experienced person; DRMs—drug resistance mutations. Fisher’s exact test was used to compare the prevalence of each DRM after 2009 with the prevalence before 2009 and to calculate odds ratio (OR). OR > 1 and OR < 1 with an adjusted *p* value < 0.05 using Holm’s method were considered increase and decrease in prevalence, respectively. Within each section mutations are ordered according to the OR of their change within treated persons. Four mutations increased in prevalence among RTI-naïve persons: K103N, V106M, G190A/E. L100I decreased in prevalence among RTI-naïve persons.

**Table 4 viruses-13-00879-t004:** Prevalence of Additional Nonpolymorphic NNRTI-Resistance Mutations in RTI-Naïve and NNRTI-Experienced Persons before and after 2009.

				<2009			≥2009		
Pos	AA	2009	2020	%Naïve (*n* = 54,910)	%Rx (*n* = 27,755)	Rx/Naïve Ratio	%Naïve (*n* = 45,940)	%Rx (*n* = 22,219)	Rx/Naïve Ratio
**DRMs that increased in prevalence in treated persons**									
227	C	2	4	0 (0.01)	0.03	9	0 (0.01)	0.12	54
179	M	1	1	0.02 (0.21)	0.13	7	0.02 (0.13)	0.27	11
179	L	0	4	0.01 (0.05)	0.17	15	0.01 (0.08)	0.32	36
138	K	2	4	0.11 (0.31)	0.44	4	0.12 (0.33)	0.75	6
138	Q	0	4	0.03 (0.17)	1.00	37	0.07 (0.20)	1.63	25
227	L	2	3	0.06 (0.18)	2.19	36	0.03 (0.04)	3.50	106
221	Y	1	4	0.08 (0.24)	6.55	79	0.08 (0.40)	8.14	103
**DRMs that did not change in prevalence in treated persons**									
236	L	2	2	0.03 (0.19)	0.09	3	0.04 (0.06)	0.18	5
234	I	1	3	0.02 (0.08)	0.29	15	0.01 (0.05)	0.43	47
101	N	2	1	0.02 (0.07)	0.50	25	0.02 (0.20)	0.65	27
238	T	2	2	0.06 (0.19)	2.73	47	0.05 (0.20)	3.02	66
190	C	3	3	0 (0)	0.13	100	0 (0)	0.14	100
190	Q	3	3	0 (0)	0.34	100	0 (0)	0.37	100
101	H	1	3	0 (0)	1.32	100	0.01 (0.02)	1.32	151
188	F	1	2	0.01 (0.03)	0.57	79	0.01 (0.02)	0.56	65
181	F	0	1	0.01 (0.02)	0.22	17	0.01 (0.08)	0.20	18
238	N	1	2	0.05 (0.19)	0.56	12	0.08 (0.19)	0.45	5
101	T	0	1	0.01 (0.02)	0.36	33	0.02 (0.03)	0.23	13

Abbreviation: Pos—amino acid position; AA—amino acid; 2009—Number of expert lists with the mutation in 2009; 2020—Number of expert lists with the mutation in 2020; %Naïve—percent prevalence among RTI-naïve persons; %Rx—percent prevalence among NNRTI-experienced persons; DRMs—drug resistance mutations. Fisher’s exact test was used to compare the prevalence of each DRM after 2009 with the prevalence before 2009 and to calculate odds ratio (OR). OR > 1 and OR < 1 with an adjusted *p* value < 0.05 using Holm’s method were considered increase and decrease in prevalence, respectively. Within each section mutations are ordered according to the OR of their change within treated persons. None of the mutations changed in prevalence among RTI-naïve persons.

**Table 5 viruses-13-00879-t005:** Prevalence of Surveillance PI-Resistance Mutations in PI-Naïve and PI-Experienced Persons before and after 2009.

				<2009			≥2009		
Pos	AA	2009	2020	%Naïve (*n* = 61,067)	%Rx (*n* = 22,266)	Rx/Naïve Ratio	%Naïve (*n* = 54,813)	%Rx (*n* = 10,534)	Rx/Naïve Ratio
**DRMs that increased in prevalence in treated persons**									
47	A	4	4	0 (0)	0.4	>>>	0 (0)	1.0	>>>
50	L	4	4	0.01 (0.03)	1.5	183	0 (0)	2.5	>>>
**DRMs that decreased in prevalence in treated persons**									
84	C	2	2	0 (0)	0.1	>>>	0 (0)	0.0	NA
84	A	3	3	0 (0)	0.1	>>>	0 (0)	0.0	NA
48	V	4	4	0 (0)	3.7	>>>	0 (0)	0.7	>>>
54	T	4	4	0.01 (0.07)	0.8	72	0.01 (0.13)	0.2	22
54	S	4	3	0 (0)	0.7	>>>	0 (0)	0.1	>>>
54	M	4	4	0 (0.14)	2.3	467	0.01 (0.01)	0.6	101
73	T	4	4	0 (0)	2.4	>>>	0 (0)	0.6	>>>
90	M	4	4	0.22 (0.30)	29.6	134	0.21 (0.42)	9.6	45
54	A	4	4	0 (0)	1.3	>>>	0 (0)	0.4	>>>
82	T	4	4	0 (0)	2.8	>>>	0 (0)	0.8	>>>
48	M	2	2	0 (0)	0.5	>>>	0 (0)	0.2	>>>
84	V	4	4	0.01 (0.07)	12.2	830	0.03 (0.06)	3.8	130
73	C	3	3	0 (0.01)	1.1	346	0.01 (0.03)	0.3	62
73	S	4	4	0.02 (0.08)	8.3	421	0.03 (0.08)	2.7	84
73	A	4	2	0 (0.03)	0.6	193	0 (0.03)	0.2	60
53	Y	3	2	0.01 (0.11)	0.4	30	0.03 (0.22)	0.2	5
46	L	4	4	0.31 (0.39)	9.2	30	0.37 (0.91)	3.4	9
30	N	4	4	0.01 (0.05)	6.2	471	0.02 (0.02)	2.4	148
82	S	4	4	0 (0.02)	1.3	392	0 (0.01)	0.5	281
47	V	4	4	0.02 (0.15)	4.5	213	0.02 (0.07)	2.0	120
24	I	4	4	0.02 (0.07)	5.6	264	0.01 (0.03)	2.5	454
85	V	3	3	0.16 (0.53)	4.6	29	0.15 (0.37)	2.1	14
54	L	4	4	0.01 (0.03)	3.0	231	0.01 (0.06)	1.4	189
54	V	4	4	0.01 (0.02)	23.1	3529	0.01 (0.03)	12.0	1101
32	I	4	4	0.01 (0.05)	5.0	337	0.01 (0.07)	2.5	458
82	A	4	4	0.04 (0.14)	22.0	612	0.03 (0.13)	12.2	352
53	L	4	4	0.03 (0.14)	5.9	179	0.02 (0.11)	3.0	126
88	D	4	4	0.02 (0.06)	5.4	219	0.02 (0.07)	2.7	167
46	I	4	4	0.26 (0.34)	21.2	82	0.34 (0.61)	12.6	37
82	F	4	4	0 (0)	1.8	1086	0.02 (0.05)	1.1	53
88	S	4	4	0.02 (0.10)	1.6	69	0 (0)	1.0	>>>
50	V	4	4	0.01 (0.05)	1.8	136	0 (0.01)	1.2	323
82	C	3	2	0 (0)	0.6	>>>	0 (0.01)	0.4	117
83	D	2	3	0.02 (0.05)	0.9	50	0.03 (0.07)	0.6	21
**DRMs that did not change in prevalence in treated persons**									
82	M	3	2	0 (0)	0.3	>>>	0 (0.13)	0.4	114
76	V	4	4	0.01 (0.14)	3.3	403	0.01 (0.03)	3.3	300
23	I	2	2	0.03 (0.03)	1.3	43	0.03 (0.05)	1.1	34
82	L	3	3	0.02 (0.04)	0.3	14	0.03 (0.06)	0.2	7

Abbreviation: Pos—amino acid position; AA—amino acid; 2009—Number of expert lists with the mutation in 2009; 2020—Number of expert lists with the mutation in 2020; %Naïve—percent prevalence among PI-naïve persons; %Rx—percent prevalence among PI-experienced persons; DRMs—drug resistance mutations. Fisher’s exact test was used to compare the prevalence of each DRM after 2009 with the prevalence before 2009 and to calculate odds ratio (OR). OR > 1 and OR < 1 with an adjusted *p* value < 0.05 using Holm’s method were considered increase and decrease in prevalence, respectively. Within each section mutations are ordered according to the OR of their change within treated persons. N88S decreased in prevalence among PI-naïve persons.

**Table 6 viruses-13-00879-t006:** Prevalence of Additional Nonpolymorphic PI-Resistance Mutations in PI-Naïve and PI-Experienced Persons before and after 2009.

				<2009			≥2009		
Pos	AA	2009	2020	%Naïve (*n* = 61,067)	%Rx (*n* = 22,266)	Rx/Naïve Ratio	%Naïve (*n* = 54,813)	%Rx (*n* = 10,534)	Rx/Naïve Ratio
**DRMs that increased in prevalence in treated persons**									
89	T	3	2	0 (0.05)	0.43	88	0.01 (0.03)	0.95	104
**DRMs that decreased in prevalence in treated persons**									
93	M	3	1	0.13 (0.55)	0.69	5	0.14 (0.39)	0.23	2
11	L	1	1	0.04 (0.14)	0.66	16	0.03 (0.20)	0.24	9
48	A	2	2	0 (0)	0.47	285	0 (0.03)	0.17	47
89	V	4	4	0.07 (0.43)	3.66	52	0.05 (0.22)	1.38	28
66	V	0	1	0.09 (0.37)	1.06	12	0.08 (0.42)	0.43	5
24	F	2	1	0.05 (0.09)	0.59	12	0.01 (0.11)	0.28	38
34	Q	1	1	0.09 (0.74)	2.62	30	0.09 (0.69)	1.31	15
71	I	4	2	0.08 (0.15)	3.24	40	0.11 (0.28)	1.69	16
43	T	3	3	0.12 (0.16)	5.62	47	0.11 (0.15)	3.01	28
71	L	3	2	0 (0.02)	0.42	129	0 (0)	0.25	>>>
10	R	3	1	0.02 (0.04)	0.42	28	0.01 (0.02)	0.25	22
74	P	3	4	0.05 (0.08)	1.91	35	0 (0.13)	1.28	351
10	F	4	4	0.22 (0.33)	8.58	38	0.23 (0.50)	6.66	29
20	T	4	3	0.1 (0.20)	5.32	55	0.09 (0.17)	4.14	47
**DRMs that did not change in prevalence in treated persons**									
41	T	1	1	0.04 (0.13)	0.09	2	0.05 (0.22)	0.11	3
88	G	1	1	0 (0)	0.17	>>>	0 (0)	0.13	>>>
46	V	1	1	0.04 (0.11)	0.53	12	0.06 (0.13)	0.45	7

Abbreviation: Pos—amino acid position; AA—amino acid; 2009—Number of expert lists with the mutation in 2009; 2020—Number of expert lists with the mutation in 2020; %Naïve—percent prevalence among PI-naïve persons; %Rx—percent prevalence among PI-experienced persons; DRMs—drug resistance mutations. Fisher’s exact test was used to compare the prevalence of each DRM after 2009 with the prevalence before 2009 and to calculate odds ratio (OR). OR > 1 and OR < 1 with an adjusted *p* value < 0.05 using Holm’s method were considered increase and decrease in prevalence, respectively. Within each section mutations are ordered according to the OR of their change within treated persons. L24F and T74P decreased in prevalence among PI-naïve persons.

## Data Availability

Virus sequences and treatment history of persons from the virus sequences were obtained available at: https://hivdb.stanford.edu/pages/geno-rx-datasets.html accessed on 8 May 2021.

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
