# Peer review of "Temporal Trends in HIV-1 Mutations Used for the Surveillance of Transmitted Drug Resistance"

_viruses, 2021, doi:10.3390/v13050879_

Round 1

Reviewer 1 Report

This is very useful research to better understand SDRM list updates and to monitor TDR. It was clearly written and easy to understand. According to the World Health Organization antiretroviral guidelines, HIV-1 integrase strand transfer inhibitors (INSTIs) are recommended as first-line therapy for HIV-infected adults. The limitation of the study is the absence of information on INSTIs-SDRMs. Therefore, authors should add data regarding candidate INSTIs-SDRMs in the article.

Minor comments:

1、Line 106: “naive” should be “naïve”.

2、Line 190: “Table 6” should be “Table 4”.

3、Tables 1-6: The decimal number should keep consistent.

4、The footnote of Tables 1-6: “%Naive” should be “%Naïve”.

5、The format of references should be checked thoroughly.

Author Response

This is very useful research to better understand SDRM list updates and to monitor TDR. It was clearly written and easy to understand. According to the World Health Organization antiretroviral guidelines, HIV-1 integrase strand transfer inhibitors (INSTIs) are recommended as first-line therapy for HIV-infected adults. The limitation of the study is the absence of information on INSTIs-SDRMs. Therefore, authors should add data regarding candidate INSTIs-SDRMs in the article.

The reviewer is correct that INSTIs have become increasingly important. However, we recently published two studies that included the information requested by the reviewer. The first was a study that classified INSTI-resistance mutations according to their presence on expert lists, their prevalence in INSTI-naïve and -experienced persons with HIV, and their association with reduced in vitro susceptibility (Tzou et al, Integrase strand transfer inhibitor (INSTI)-resistance mutations for the surveillance of transmitted HIV-1 drug resistance, PMID 31617907). We used these data to develop a list of suggested mutations for INSTI-TDR surveillance. The second study examined the utility of the list of INSTI-associated DRMs using data from studies published since 2010 (Bailey et al, Integrase Strand Transfer Inhibitor Resistance in Integrase Strand Transfer Inhibitor-Naive Persons, PMID 33683148). We have therefore modified the paper as follows:

In the Introduction, we have added the following sentence (Lines 41-45): “In a recent study, we and others reported the prevalence of integrase strand transfer inhibitor (INSTI)-resistance associated mutations in INSTI-naïve and INSTI-experienced persons and developed a list of nonpolymorphic INSTI-selected DRMs that could be useful for the surveillance of INSTI-TDR [3].”

Minor comments:

  1. Line 106: “naive” should be “naïve”.

The text has been changed (now in Line 113).

  1. Line 190: “Table 6” should be “Table 4”.

The table number has been corrected (now in Line 224). 

  1. Tables 1-6: The decimal number should keep consistent.

The decimal numbers have been corrected.

  1. The footnote of Tables 1-6: “%Naive” should be “%Naïve”.

The footnotes have been changed.

  1. The format of references should be checked thoroughly.

The format of references have been checked and corrected according to the journal guideline.

Reviewer 2 Report

Manuscript 1195520 provides a timely summary of HIV-1 drug resistance mutations (TRMs) that may offer new insights into the evolving landscape of transmitted drug resistance in the new era of universal therapy. The sample sizes are quite impressive, but the analytical methods only scratch the surface of the real picture.

Main concerns

  1. The list of TRMs compiled in 2009 is apparently outdated, which justifies a reappraisal of TRMs in various clinical settings now. However, comparing TRMs pre- and post-2009 is not sufficient enough to capture the “temporal changes” over time, which is the topic of this paper. A more meaningful approach, which requires some in-depth evaluation, is to plot the changes over time based on calendar dates (yearly or otherwise) as permitted by the sample size.
  1. Changes in the guideline for applying antiretroviral (ARVs) took a major turn in 2012 when treatment as prevention was rolled out. It should be very useful to compare the temporal changes in TRM prevalence before and after 2012. In other words, universal treatment may have changed the dynamics of TRMs and transmitted drug resistance.
  1. One recurrent problem in this paper is description of changes in certain TRMs without referring to the magnitude of differences (e.g., proportions or ratios), confidence intervals or their statistical significance.
  1. Temporal changes are better captured by graphs if the data can be broken down by calendar dates.

Minor issues

  • It would be nice to have a graph showing the mapping of key drug resistance mutations along the HIV-1 proteome. The prevalence of these mutations can be color-coded (as in heatmaps) as well.
  • When specific drug resistance mutations are presented in the text, a brief reference to the actual viral protein should help with the orientation, especially for readers who are not familiar with the division of three classes of ARVs.
  • Analyses of HIV-1 sequences often require several layers of quality assurance. For this study, it will be important to ensure that sequences corresponding to non-competent viral genomes are excluded.

Author Response

Manuscript 1195520 provides a timely summary of HIV-1 drug resistance mutations (TRMs) that may offer new insights into the evolving landscape of transmitted drug resistance in the new era of universal therapy. The sample sizes are quite impressive, but the analytical methods only scratch the surface of the real picture.

Main concerns

  1. The list of TRMs compiled in 2009 is apparently outdated, which justifies a reappraisal of TRMs in various clinical settings now. However, comparing TRMs pre- and post-2009 is not sufficient enough to capture the “temporal changes” over time, which is the topic of this paper. A more meaningful approach, which requires some in-depth evaluation, is to plot the changes over time based on calendar dates (yearly or otherwise) as permitted by the sample size.

We have created two additional multi-part supplementary figures that display the temporal changes in mutation prevalence by year. Figure S1 contains three figures, one for each drug class displaying the SDRM prevalence in drug-class experienced persons. Figure S2 contains three figures, one for each drug class displaying the candidate SDRM prevalence in drug-class experienced persons. In each figure, the plotted lines are colored according to whether the mutation displayed an increase or decrease in prevalence.

We refer to these figures in the text as indicated below:

Lines 141-142: “Figure S1 (A) displays the yearly prevalence of each NRTI-associated SDRM in NRTI-experienced persons.”

Lines 197-198: “Figure S1 (B) displays the yearly prevalence of each NNRTI-associated SDRM in NNRTI-experienced persons.”

Lines 258-259: “Figure S1 (C) displays the yearly prevalence of each PI-associated SDRM in PI-experienced persons.”

Lines 149-151: “Figure S2 (A) displays the yearly prevalence of each NRTI-associated candidate SDRM in NRTI-experienced persons.”

Lines 210-212: “Figure S2 (B) displays the yearly prevalence of each NNRTI-associated candidate SDRM in NNRTI-experienced persons.”

Lines 270-272: “Figure S2 (C) displays the yearly prevalence of each PI-associated candidate SDRM in PI-experienced persons.”

  1. Changes in the guideline for applying antiretroviral (ARVs) took a major turn in 2012 when treatment as prevention was rolled out. It should be very useful to compare the temporal changes in TRM prevalence before and after 2012. In other words, universal treatment may have changed the dynamics of TRMs and transmitted drug resistance.

This study was not designed to track temporal TDR rates by region. However, we recently published a study that analyzed temporal TDR rates (Rhee et al, HIV-1 transmitted drug resistance surveillance: shifting trends in study design and prevalence estimates, PMID 32936523). In that study, we also bisected the sequence data into two periods, 2009-2011 (n=19,440) and 2012-2018 (n=20,223). We reported the significant temporal changes in each region between the two time periods in overall-, NRTI-, NNRTI- and PI-TDR as well as individual mutations. As in our current study, the candidate SDRMs were much less common than the established SDRMs.

However, in this study, we chose the 2009 date to divide the dataset because, we pre-2009 prevalence data in ART-naïve and ART-experienced was used to develop the list of SDRMs and we focused primarily on the prevalence of mutations in ART-experienced persons as changes in this population will be observed before the changes are reflected in ART-naïve persons.

  1. One recurrent problem in this paper is description of changes in certain TRMs without referring to the magnitude of differences (e.g., proportions or ratios), confidence intervals or their statistical significance.

We revised the Method’s section to clarify how mutation prevalences were compared and to indicate that we did quantify the magnitude of changes to mutation prevalences. We now provide the odds ratios and adjusted p values for the mutation prevalence changes in supplementary tables S6, S7, and S8.

Revised description of statistical approach in Methods (Lines 113-118): “To quantify the change in prevalence of DRMs before and after 2009, we compared the prevalence of each DRM after 2009 with the prevalence before 2009 using Fisher’s exact test. P values were adjusted using Holm’s method to control the family-wise error rate for multiple hypothesis testing. The changes in the prevalence with an adjusted p values <0.05 were considered significant and ranked according to the odds ratios (ORs).”

We refer to the tables S6, S7 and S8 in the text as indicated below:

Lines 151-153: “Table S6 contains the ORs and adjusted p values comparing the prevalence of each NRTI-associated SDRM and candidate SDRM before and after 2009 in RTI-naïve and NRTI-experienced persons.”

Lines 212-214: “Table S7 contains the ORs and adjusted p values comparing the prevalence of each NNRTI-associated SDRM and candidate SDRM before and after 2009 in RTI-naïve and NNRTI-experienced persons.”

Lines 272-274: “Table S8 contains the ORs and adjusted p values comparing the prevalence of each PI-associated SDRM and candidate SDRM before and after 2009 in PI-naïve and PI-experienced persons”. 

  1. Temporal changes are better captured by graphs if the data can be broken down by calendar dates.

We have addressed this in Figures S1 and S2 as described in the response to the first critique.

Minor issues

  • It would be nice to have a graph showing the mapping of key drug resistance mutations along the HIV-1 proteome. The prevalence of these mutations can be color-coded (as in heatmaps) as well.

We have added structural figures (Figure 1-3) in which individual SDRMs and candidate SDRMs have been color-coded according to their change in prevalence in ART-experienced persons and their priority for possible inclusion in an updated list of SDRMs.

We refer to the Figure 1-3 in the text as indicated below:

Lines 163-165: “Figure 1 shows the locations of the NRTI-SDRMs and the six candidate mutations (K65N, T69deletion and K70G/N/Q/T) within the three-dimensional structure of the polymerase coding region of p66 HIV-1 RT.”

Lines 220-223: “Figure 2 shows the locations of the NNRTI-SDRMs and the six candidate mutations (E138K/Q, V179L, H221Y and F227C/L) within the three-dimensional structure of the polymerase coding region of p66 and the finger domain of p51 HIV-1 RT.”

Lines 282-284: “Figure 3 shows the locations of the PI-SDRMs and the three candidate mutations (L10F, T74P and L89V) within the three-dimensional structure of HIV-1 protease.”

  • When specific drug resistance mutations are presented in the text, a brief reference to the actual viral protein should help with the orientation, especially for readers who are not familiar with the division of three classes of ARVs. Structural figures are added as described above.
  • Analyses of HIV-1 sequences often require several layers of quality assurance. For this study, it will be important to ensure that sequences corresponding to non-competent viral genomes are excluded.

We added our description of our quality control filter to the Methods section (Lines 91-93): “Quality control involved restricting our analysis to sequences generated by direct PCR dideoxy-nucleotide sequencing from plasma samples that displayed no evidence for G-to-A hypermutation or an excess of highly unusual mutation or stop codons.”

Reviewer 3 Report

The submitted article, “Temporal Trends in HIV-1 Mutations Used for the Surveillance of Transmitted Drug Resistance,” by Rhee et al., is an expansion of a compendium of work by this group on considerations for identifying and monitoring mutations associated with HIV resistance to antiretroviral drugs. With newer, more potent regimens the interest in transmitted resistance may have waned with the perception that current ARVs are likely to still be active against these variants. Indeed, at this point in HIV treatment this is largely true; however, it is the view of this reviewer that “honeymoon” periods of potent suppression coincide with the introduction of new ARVs, but those periods are not indefinite. Moreover, in countries with limited treatment options the consequence of TDRMs can be weightier. Therefore, having a systematic way of monitoring for resistance trends, particularly of multi-drug resistance, is an important safety net to preserve therapeutic potency and effectiveness of PrEP. The work presented here compiles current evidence of mutations that appear to be transmitted at frequencies above what is seen with natural polymorphisms. The inclusion of all subtypes as candidate mutations of interest is helpful for a global approach, as would be a concern for the WHO. However, some additional candidate mutations may be considered for country-level surveillance in that high polymorphic backgrounds may be ignored with certain subtypes if those genotypes are not observed to circulate in those regions.  Moreover, if a particular class of TDRMs might be more significant for a given region, that could be highlighted.

Specific comments:

  1. The authors should discuss why integrase inhibitor mutations are not considered/discussed in this paper. While their prevalence might not be substantial at this time, given the importance of this ARV class their attention is warranted.
  2. Lines 90-92. If sequences containing two or more mutations from the pre-2007 list did not include 215/219 TAM NRTI mutations, then they might still be included on this list. The circumstances may have naturally evolved in in the accumulation of non-215/219 NRTIs, but it would be informative to know that the authors had considered this given those TDRMs would still be relevant today.
  3. Lines 97-98. The wording, “Nonpolymorphic DRMs were then defined as… below 1.0% in each of the eight major subtypes and circulating recombinant forms (CRFs):” is unclear. Are the authors intending to say the CRFs are also from ARV-naïve persons?
  4. Line 110. Here, please elaborate on what constitutes the “10 sets of sequences.”
  5. A couple typos: Line 132, four of the T215 revertants “have”; line 272, have not “been” linked.

Author Response

The submitted article, “Temporal Trends in HIV-1 Mutations Used for the Surveillance of Transmitted Drug Resistance,” by Rhee et al., is an expansion of a compendium of work by this group on considerations for identifying and monitoring mutations associated with HIV resistance to antiretroviral drugs. With newer, more potent regimens the interest in transmitted resistance may have waned with the perception that current ARVs are likely to still be active against these variants. Indeed, at this point in HIV treatment this is largely true; however, it is the view of this reviewer that “honeymoon” periods of potent suppression coincide with the introduction of new ARVs, but those periods are not indefinite. Moreover, in countries with limited treatment options the consequence of TDRMs can be weightier. Therefore, having a systematic way of monitoring for resistance trends, particularly of multi-drug resistance, is an important safety net to preserve therapeutic potency and effectiveness of PrEP. The work presented here compiles current evidence of mutations that appear to be transmitted at frequencies above what is seen with natural polymorphisms. The inclusion of all subtypes as candidate mutations of interest is helpful for a global approach, as would be a concern for the WHO. However, some additional candidate mutations may be considered for country-level surveillance in that high polymorphic backgrounds may be ignored with certain subtypes if those genotypes are not observed to circulate in those regions.  Moreover, if a particular class of TDRMs might be more significant for a given region, that could be highlighted.

Specific comments:

  1. The authors should discuss why integrase inhibitor mutations are not considered/discussed in this paper. While their prevalence might not be substantial at this time, given the importance of this ARV class their attention is warranted. We have addressed this comment in the response to the main comment of Reviewer 1.
  1. Lines 90-92. If sequences containing two or more mutations from the pre-2007 list did not include 215/219 TAM NRTI mutations, then they might still be included on this list. The circumstances may have naturally evolved in in the accumulation of non-215/219 NRTIs, but it would be informative to know that the authors had considered this given those TDRMs would still be relevant today. The approach we used to exclude sequences that likely reflect TDR from the ART-naïve category is independent of whether the 2009 or very similar 2007 list were used.
  1. Lines 97-98. The wording, “Nonpolymorphic DRMs were then defined as… below 1.0% in each of the eight major subtypes and circulating recombinant forms (CRFs):” is unclear. Are the authors intending to say the CRFs are also from ARV-naïve persons? We have added the phrase “ART-class naïve persons” at the end of the sentence to indicate unequivocally that the 1% threshold for sequences belonging to the eight major subtypes and CRFs also referred to ART-class naïve persons (now in Line 106).
  1. Line 110. Here, please elaborate on what constitutes the “10 sets of sequences.”

Original sentence: “To evaluate the potential impact of adding mutations to the SDRM list, we selected 10 sets of sequences from ART-naïve persons in recently published TDR studies containing ³1000 persons [5].”

Revised sentence (now in Lines 118-120): “To evaluate the potential impact of adding mutations to the SDRM list, we selected recently published TDR studies that contained sequences from ³1,000 ART-naïve persons (Table S5) [6]

  1. A couple typos: Line 132, four of the T215 revertants “have”; line 272, have not “been” linked.

The text in line 132 (now in Line 140) and 272 (now in Line 336) have been corrected.

Round 2

Reviewer 2 Report

The edits and improvements to this manuscript are impressive.